



# Three-stream modelling of radiative transfer for the simulation of Black Sea biogeochemistry in a NEMO framework

Loïc Macé[12], Luc Vandenbulcke[1], Jean-Michel Brankart[2], Jean-François Grailet[1], Pierre Brasseur[2], and Marilaure Grégoire[1]

[1]MAST-FOCUS research group, Department of Astrophysics, Geophysics and Oceanography, University of Liège, Belgium
[2]Univ. Grenoble Alpes, CNRS, INRAE, IRD, Grenoble INP, IGE, Grenoble, France

**Correspondence:** Loïc Macé (loic.mace@uliege.be)

**Abstract.**

In this paper, we propose a three-stream ocean radiative transfer (RT) module as an extension of the NEMO ocean modelling framework. This module solves the subsurface irradiance field in 1D water columns, discriminating between two downward streams, direct and scattered, and a backscattered upward stream. The module solves 33 wavebands ranging between 250 and 4000 nm, with a finer 25 nm resolution in the visible range. The sea surface reflectance is also calculated as a model output, based on the ratio between the upward and downward irradiances at the air-sea interface. An optional feedback towards NEMO is presented, enabling the use of irradiances to compute temperature in the hydrodynamics. The module also includes a stochastic version in which the inherent optical properties of the main optically active components of seawater can be perturbed. This mode is meant to account for uncertainty in the modelling of marine optics. This module is can be plugged to any NEMO configuration, with the computation of optical properties either driven by a biogeochemical model or directly forced into the RT module.

We apply this module in a test case for the Black Sea, coupled with the physical-biogeochemical framework NEMO 4.2.0-BAMHBI. We find that substituting the existing radiative transfer scheme with our model unlocks the ability to simulate radiometric variables that can be compared more truthfully to observations, both *in situ* and from remote-sensing. We also find that using irradiances to compute the temperature and PAR in the model maintains consistency in the calculation of physical and biogeochemical variables in the model, such as temperature or chlorophyll concentration, while enabling additional capabilities in the model in the simulation of radiometric quantities.

## 1 Introduction

The abundance of light is a driving factor in the development of marine ecosystems. First, the spectral composition of solar radiation has a direct influence on the abundance and species composition of phytoplankton through photosynthesis, which converts inorganic compounds into organic material (Mobley et al., 2015). Solar radiation is also a driver of the evolution of sea temperature (Cahill et al., 2023). As such, it influences the vertical profiles of the temperature and the depth of the





thermocline. Light also has consequences on the $N_2O$ inventory (Berthet et al., 2023), and affects some chemical reactions,
such as nitrification, which are inhibited under high irradiance (Horrigan et al., 1981; Yang et al., 2022). Thus, light plays a
complex but key role in ecosystem dynamics, and a good representation of its propagation and spectral composition is essential
to predict marine ecosystems (Patara et al., 2012; Xiu and Chai, 2014; Skákala et al., 2022).

Yet, the representation of light in marine ecosystem models is still often oversimplified. In most cases, only the direct
light stream is represented, whereas the diffuse part is not, with only a few wavebands that are considered (typically two
to four). Over time, radiative transfer (RT) models of increasing complexity and accuracy have been coupled with physical-
biogeochemical models. This effort has been initiated by Aas (1987); Gregg and Carder (1990); Ackleson et al. (1994) with
models that solve the propagation of three streams of irradiance in the vertical direction: the direct downward irradiance and
the diffuse (upward and downward) irradiances. RT models have been increasingly used in various applications in 1D (Bissett
et al., 1999; Terzić et al., 2021) and 3D configurations (Dutkiewicz et al., 2015; Baird et al., 2016; Gregg and Rousseaux, 2017;
Álvarez et al., 2022). These applications have demonstrated the benefits of coupling RT models with biogeochemical models,
improving the representation of the ambient light field (Fujii et al., 2007), primary production (Kettle and Merchant, 2008),
and phytoplankton communities (Gregg and Rousseaux, 2016). Recent configurations solve the spectral composition of light
to reach resolutions up to 25 nm in the visible range (Lazzari et al., 2021), and the increase in spectral resolution allows for a
more precise calculation of heating rates in the upper ocean (Morel, 1988).

The vertical propagation of the three irradiance streams, their intensity, spectral composition, and the sea surface reflectance
are governed by the Inherent Optical Properties (IOPs) of the seawater. IOPs are determined by the composition of the water
column, in particular in optically active water components. The main optically important materials are pure seawater, phy-
toplankton, mineral and organic particles, and coloured dissolved organic matter (CDOM) (Mobley, 2022). Solving the RT
equations requires information on IOPs. They are usually provided either as forcing functions or taken from the outputs of
a biogeochemical model. Advanced RT solutions have been developed and commercialised over the years, such as the Hy-
drolight/Ecolight software first introduced in Mobley (1989) or the OSOAA model (Chami et al., 2015), which even includes
light polarisation. These models are widely documented and used, and typically use observations as inputs and are not plugged
with ecosystem models. These models are also computationally expensive, and less complex models are typically used to
solve RT in ecosystem models. Perhaps the most significant effort to couple a three-stream RT model to a coupled physical-
biogeochemical model is described in Dutkiewicz et al. (2015), with the MITgcm-DARWIN configuration. Nonetheless, an
important calibration and validation effort is required to switch to a different physical or biogeochemical model.

In recent decades, the amount of radiometric data has largely increased thanks to the development of optical sensors used on
satellite and autonomous platforms such as the Argo network (De Nodrest et al., 2022; Begouen Demeaux and Boss, 2022).
Information on the spectral irradiance and reflectance is now available at high spatial, temporal, and spectral resolution. How-
ever, spectral radiometric quantities are usually not simulated by biogeochemical models, and simulations and data have to be





connected to observations through inversion models that retrieve biogeochemical model variables from the radiometric observations. The most common example is the sea surface chlorophyll that is derived from remote-sensing sea surface reflectance using methods such as band-ratio algorithms or machine learning. When coupled with biogeochemical models, RT models can serve as an observation operator connecting dynamically the simulated biogeochemical variables with spectral radiative quantities, providing a relevant framework for better use of radiometric data (Dutkiewicz et al., 2018).

In this paper, we propose and describe a RT module ready to be coupled with NEMO, adapted from the model described in Dutkiewicz et al. (2015). NEMO is among the most commonly used models for simulating ocean physics. Throughout the years, the representation of the light penetration in NEMO has been improved. Initially, a two-band model that solves light attenuation in the visible and near-infrared ranges was implemented, following the formulations of Paulson and Simpson (1977, 1981). Lengaigne et al. (2007) proposed to add a third band, separating the visible range between blue (400-500 nm), green (500-600 nm) and red (600-700 nm), as applied in the PISCES model (Aumont et al., 2015). This model was used in Ciavatta et al. (2014) and Hordoir et al. (2019) among other applications. A fourth band can be added in the infrared range as in Vichi et al. (2015), while still using the general formulation of Lengaigne et al. (2007). More recently, NEMO 5 included a fifth band in the ultraviolet range (Madec and the NEMO System Team, 2022), described following the formulation of Morel and Maritorena (2001). Most of these models are solving a single downward stream of irradiance, with the computation of an attenuation coefficient to dim the surface irradiance stream. Some implementations use a finer spectral resolution (Skákala et al., 2020), but the inclusion of an online coupling between NEMO and a three-stream RT model remains open to new developments, in particular in 3D configurations with high spectral resolution. This RT model is applied here to the Black Sea, presented as an additional module of the BiogeochemicAl Module for Hypoxic and Benthic influenced areas (BAMHBI Grégoire et al., 2008; Grégoire et al., 2025; Grégoire and Soetart, 2010; Capet et al., 2016). In NEMO-BAMHBI, Capet (2014) introduced the use of a rather simple three-band model. This initial optical model accounts for the properties of water and phytoplankton, with an additional correction to account for CDOM concentration and the optional inclusion of suspended minerals. The aim of this work is to improve this initial resolution of RT by replacing it by a three-stream model.

The output of this paper is a version of the three-stream RT module compatible with NEMO, including a stochastic mode, and an upgraded version of BAMHBI that includes this RT module as an additional module. The stochastic version of the module can be used to generate ensembles that account for modelling uncertainties and their quantification, similarly to what is done when quantifying uncertainties in observations, as already demonstrated in Macé et al. (2025). A 1D testing version of the RT model is also provided as a convenient way to run it for a single water column and tune its parameters with, for instance, radiometric data from BGC-Argo floats. The RT module is tested and validated in a regional configuration of NEMO for the Black Sea. In particular, the IOPs and model parameters were calibrated and tested to adjust the initial model to our test case. An additional objective of this study is to evaluate the consequences of substituting the RT scheme from the current BAMHBI configuration and to evaluate the possibilities that are now offered with the updated coupled model. The RT module described here is integrated into NEMO and requires the IOPs that can be provided either by a biogeochemical model or by forcing





functions. The use of the RT module is demonstrated in the Black Sea using the well-tested NEMO-BAMHBI configuration.


The description of the RT module and its integration within the NEMO framework are presented in Sect. 2. In Sect. 3, we focus on the application of this model to the modelling of Black Sea ecosystems, with an upgraded version of the BAMHBI biogeochemical model. In Sect. 4, we analyse the consequences of using the fully coupled framework and the additions brought by the RT module, with several application cases in the Black Sea. The impact of using the RT module is assessed by com-

paring its results in terms of simulated physics and biogeochemistry with those obtained using the three-band model currently available with BAMHBI. We validate the simulated sea surface reflectance with both *in situ* and remote-sensing data. We also perform a comparison of surface chlorophyll using the inversion algorithms that are traditionally used to process remote-sensing reflectance data on the variables of our model. Finally, Sect. 5 discusses model performances and limitations along with a discussion of the perspectives provided by this implementation.

## 2 The radiative transfer module

This section describes the equations and the code of the RT model, which are adapted from Dutkiewicz et al. (2015) in which they were presented for the global MITgcm-Darwin configuration. The model is adapted here for its coupling with NEMO and a biogeochemical model. The formulation of the model is influenced by the BAMHBI model that will be presented in the next section with the test case. We also detail the IOPs that are considered in this formulation and the stochastic version of the

model.

### 2.1 Radiative transfer modelling

The RT model considers light absorption and scattering in two directions, downward and upward. It solves three streams of irradiance. Two streams are downward: the direct stream $E_d$ represents light whose path has not been affected by the scattering (forward or backward) and absorption of optical active components, and the scattered stream $E_s$ represents light that has been

scattered in the downward direction. The upward stream $E_u$ represents light that has been backscattered, i.e. scattered in the upward direction. The model then runs independently for each wavelength. The set of equations describing the propagation of irradiance in the water column is as follows.

$$\frac{dE_d(\lambda, z)}{dz} = -\frac{a(\lambda, z) + b_f(\lambda, z) + b_b(\lambda, z)}{\overline{\mu_d}} E_d(\lambda, z) \tag{1}$$

$$\frac{dE_s(\lambda, z)}{dz} = -\frac{a(\lambda, z) + r_s b_b(\lambda, z)}{\overline{\mu_s}} E_s(\lambda, z) + \frac{r_u b_b(\lambda, z)}{\overline{\mu_u}} E_u(\lambda, z) + \frac{b_f(\lambda, z)}{\overline{\mu_d}} E_d(\lambda, z) \tag{2}$$

$$-\frac{dE_u(\lambda, z)}{dz} = -\frac{a(\lambda, z) + r_u b_b(\lambda, z)}{\overline{\mu_u}} E_u(\lambda, z) + \frac{r_s b_b(\lambda, z)}{\overline{\mu_s}} E_s(\lambda, z) + \frac{b_b(\lambda, z)}{\overline{\mu_d}} E_d(\lambda, z) \tag{3}$$

With $a$, $b_f$ and $b_b$ being respectively the absorption, forward scattering, and backscattering coefficients, in units of $m^{-1}$, and described in more detail in section 2.3. $r_s$ and $r_u$ are dimensionless coefficients that describe the repartition of scattering





between the upward and downward directions. $\overline{\mu_d}$, $\overline{\mu_s}$, $\overline{\mu_u}$ account for the angular distribution of light in the form of average cosines, also dimensionless (Aas, 1987). By considering these parameters as constant, the system can be reduced to a tridiago-

nal system. The full description of the resolution is described in the Appendix of Dutkiewicz et al. (2015).

This system of equations is closed by two surface boundary conditions $E_{d0}$ and $E_{s0}$ respectively for the surface values of the $E_d$ and $E_s$ streams. The boundary conditions must be provided for all wavelengths solved by the RT model. The spectral resolution of the model can be adjusted freely as long as surface boundary conditions and profiles of optical properties can be

provided over the whole spectral range. At the bottom, the system is closed by assuming that there is no bottom reflection, which means that the $E_u$ stream is 0 at the bottom of the water column. This has no consequence in deep waters, but this feature may be questioned for coastal applications.

$$E_d(\lambda, z = 0) = E_{d0} \tag{4}$$

$$E_s(\lambda, z = 0) = E_{s0} \tag{5}$$

$$E_u(\lambda, z = depth) = 0 \tag{6}$$

From irradiance streams, we derive the reflectance $R$ as the ratio between the upward and downward irradiance in the upper layer of the model (i.e. below sea surface) as in the following equation.

$$R^{below}(\lambda) = \frac{E_u(\lambda, z = 0)}{E_d(\lambda, z = 0) + E_s(\lambda, z = 0)} \tag{7}$$

In order to convert the sea surface reflectance $R$ into a variable comparable to remote-sensing reflectance, as observed by

earth observation satellites, its value is corrected by accounting for the bidirectional reflectance distribution function $Q$. This coefficient depends on parameters such as the refraction index at the air/sea interface, the solar zenith angle, and the wave state of the ocean surface (Morel et al., 2002). It is set as a constant in this model.

$$R^{below}_{RS}(\lambda) = \frac{R^{below}(\lambda)}{Q} \tag{8}$$

Finally, the below-surface remote-sensing reflectance is converted into an above-surface quantity following Lee et al. (2002).

$$R_{RS}(\lambda) = \frac{T R^{below}_{RS}(\lambda)}{1 - \gamma Q R^{below}_{RS}(\lambda)} \tag{9}$$

With $T$ the radiance transmittance of the interface and $\gamma$ the internal reflection coefficient (Bai et al., 2020). $R_{RS}$ is the quantity that is then comparable to remote-sensing data.





The RT model described in this section can be used as an observation operator for the physical-biogeochemical framework, as it serves as a link between the modelling framework and the observations. We talk then of a one-way configuration, when information from the RT module is not used by the other components of the modelling framework. However, the RT module is two-way coupled if the simulated irradiances are used in the simulation of temperature and primary production in the coupled model. In this article, we only consider the two-way configuration of the coupled model.

## 2.2 Integration within the NEMO framework

The total scalar irradiance budget $E_{tot}$ is computed by the RT module over its entire spectral range and is used as the source term of Eq. 10 which describes the evolution of the heat budget. In the classic NEMO radiative schemes, $E_{tot}$ is derived only from the downward stream of irradiance computed over two to four wavebands. In this formulation, the three streams of irradiance appear in the equation and the amount of wavebands is increased due to the finer spectral resolution, as in Eq. 11.

$$\frac{\partial T}{\partial t} = \frac{\eta_H}{c_p \rho_0} \frac{\partial E_{tot}}{\partial z} \tag{10}$$

$$E_{tot} = \sum_\lambda \left( \frac{E_d(\lambda)}{\mu_d} + \frac{E_s(\lambda)}{\mu_s} - \frac{E_u(\lambda)}{\mu_u} \right) \tag{11}$$

with $c_p$ the heat capacity of water, $\rho_0$ the density of water and $\eta_H$ a constant dimensionless tuning factor. $\Delta E_d(\lambda)$, $\Delta E_s(\lambda)$ and $\Delta E_u(\lambda)$ respectively represent the difference in irradiance between the top and bottom of the model grid cell. The parameter $\eta_H$ is meant to account for the fraction of irradiance loss that is actually used to heat water. In reality, some of the energy that is lost as light propagates in seawater is used by phytoplankton for photosynthesis or to degrade particles and minerals (Del Vecchio and Blough, 2002). In most cases, the $\eta_H$ parameter should be set close to 1, and should be calibrated accordingly with observations for each use case. This temperature feedback is proposed as an additional scheme for RT that can be used in place of the default schemes defined in the `traqsr.F90` source file (e.g. two- or three-band models). This scheme is activated by setting the flag `ln_qsr_RT = .true.` in the `namtra_qsr` namelist. The $\eta_H$ parameter is to be set in the `nam_RADTRANS` namelist.

## 2.3 Inherent Optical Properties

Solving Eqs. 1 to 3 requires information on the profiles of seawater IOPs: coefficients of spectral absorption $a$, forward scattering $b_f$, and backscattering $b_b$. These variables are forced into the model or estimated by a biogeochemical model plugged with NEMO. In the three-stream RT model, we account for four optically active constituents that contribute to absorption and scattering: pure water, phytoplankton, detritic non-algal particles, and coloured dissolved organic matter (CDOM). We assume that the latter only contributes to absorption (Dutkiewicz et al., 2015; Álvarez et al., 2023). IOPs are derived from the sum of optical properties of seawater constituents that contribute to absorption and scattering.





$$a(\lambda, z) = a_w(\lambda, z) + a_{phy}(\lambda, z) + a_{prt}(\lambda, z) + a_{cdom}(\lambda, z) \tag{12}$$

$$b_f(\lambda, z) = b_{f,w}(\lambda, z) + b_{f,phy}(\lambda, z) + b_{f,prt}(\lambda, z) \tag{13}$$

$$b_b(\lambda, z) = b_{b,w}(\lambda, z) + b_{b,phy}(\lambda, z) + b_{b,prt}(\lambda, z) \tag{14}$$

Many studies have focused on the optical properties of pure water ($a_w$, $b_{f,w}$, and $b_{b,w}$), and these are now well documented (Pope and Fry, 1997; Morel et al., 2007; Mason et al., 2016). Water has a high absorbing power, except in the visible range in which light is more easily transmitted. We consider isotropic scattering of water as in Gregg and Rousseaux (2016), with equal scattering and backscattering. Absorption and scattering spectra are provided in Fig. 1.

The absorption and scattering by phytoplankton ($a_{phy}$, $b_{f,phy}$, and $b_{b,phy}$) are the sum of the contributions of the phytoplankton species that are considered, as in Eqs. 15 to 17. In biogeochemical models, phytoplankton species are typically grouped in phytoplankton functional types (PFTs) based on common traits and affinities for light or nutrients. For each PFT, absorption and scattering are computed from phytoplankton concentration and reference spectra. The number of PFTs considered in the model can be adjusted depending on the application of the model. Examples of spectra for PFTs that are considered in the use case of this article are presented in Fig. 1.

$$a_{phy}(\lambda, z) = \sum_i a_{phy}^i(\lambda) CHL^i(z) \tag{15}$$

$$b_{f,phy}(\lambda, z) = \sum_i b_{f,phy}^i(\lambda) C_{chl}^i(z) \tag{16}$$

$$b_{b,phy}(\lambda, z) = \sum_i b_{b,phy}^i(\lambda) C_{chl}^i(z) \tag{17}$$

Where $a_{phy}^i$ is the reference absorption in units of m$^2$ mgChl$^{-1}$, $b_{f,phy}^i$ and $b_{b,phy}^i$ are the reference scattering in units of m$^2$ mmolC$^{-1}$, $CHL^i$ is the chlorophyll concentration for each PFT in units of mgChl m$^{-3}$ and $C_{chl}^i$ is the carbon content in each PFT in units of mmolC m$^{-3}$.

The optical properties of non-algal particles ($a_{prt}$, $b_{f,prt}$, and $b_{b,prt}$) are derived from the concentration of particulate organic carbon (POC) as in Eqs. 18 to 20. It acts as a proxy for particle concentration, under the assumption that the particles are uniform in size and shape (Dutkiewicz et al., 2015). Reference coefficients ($a_{prt}^0$, $b_{f,prt}^0$, and $b_{b,prt}^0$) are derived from Gallegos et al. (2011) and Álvarez et al. (2022) and their spectra are provided in Fig. 1.

$$a_{prt}(\lambda, z) = a_{prt}^0(\lambda) POC(z) \tag{18}$$

$$b_{f,prt}(\lambda, z) = b_{f,prt}^0(\lambda) POC(z) \tag{19}$$

$$b_{b,prt}(\lambda, z) = b_{b,prt}^0(\lambda) POC(z) \tag{20}$$





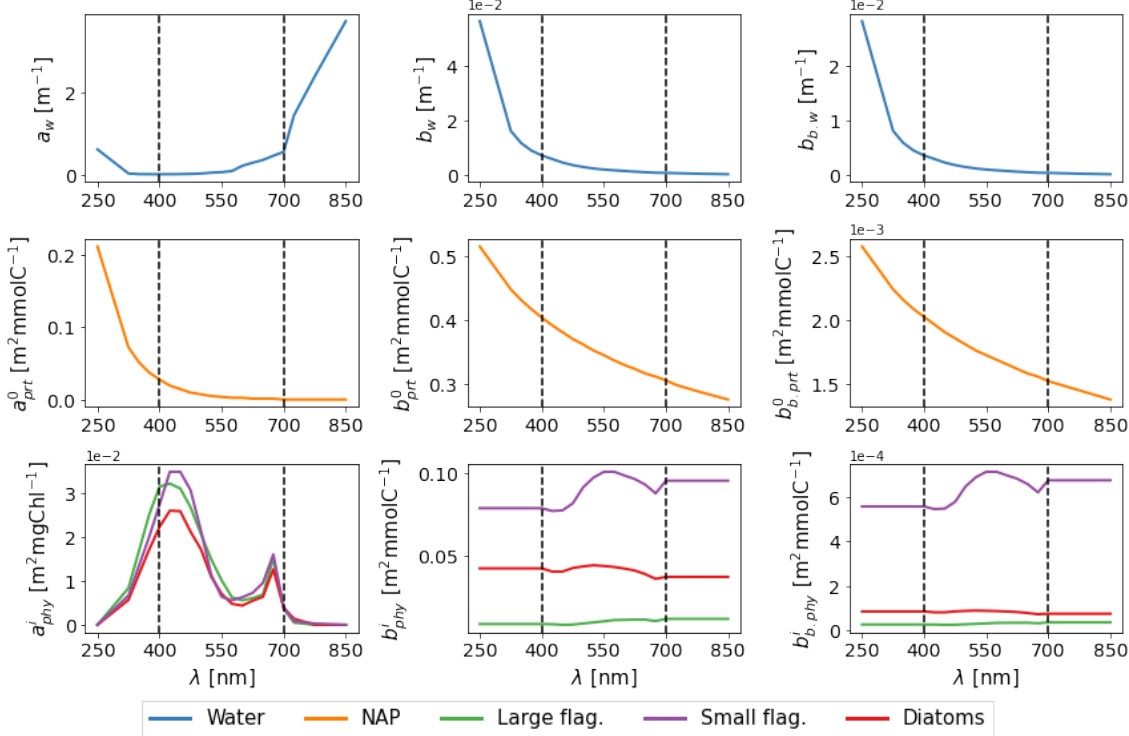

**Figure 1.** Optical properties for water (top), non-algal particles (centre) and the three PFTs defined in BAMHBI (bottom).

Absorption by CDOM ($a_{cdom}$) is calculated from a reference profile at a prescribed wavelength. We choose the 412 nm wavelength as reference here as data for irradiance streams are often available in this waveband. The reference profile can either be forced into the model of derived from an estimation of CDOM concentration provided by a biogeochemical model.

210 CDOM absorption is then propagated in other wavebands with the following exponential law.

$$a_{cdom}(\lambda) = a_{cdom}(\lambda_{ref})e^{-S_{cdom}(\lambda - \lambda_{ref})} \tag{21}$$

With $a_{cdom}$ the reference forcing profile at $\lambda_{ref} = 412$ nm and $S_{cdom}$ the slope factor describing the exponential decrease in absorption in longer wavebands (Twardowski et al., 2004; Kitidis et al., 2006; Dutkiewicz et al., 2015). Thus CDOM principally absorbs in short wavelengths. The values for $s_{cdom}$ and all the other parameters of the RT model need to be specified by the

215 user depending on the application case of the model.

### 2.4 Stochastic version of the radiative transfer model

In addition to the deterministic version of the RT module, we provide a stochastic version of the RT model, that offers the possibility to consider uncertainties in the modelling of IOPs. The stochastic version of the model is based on the introduction of



stochastic perturbations in the parameterisation of the IOPs, based on the generic approach developed by Brankart et al. (2015).
It was presented in Macé et al. (2025) where the RT module was used in a one-way coupled configuration (i.e., as an observation operator). Perturbations in the IOPs are meant to account both for uncertainties in the specific coefficients for absorption and scattering (as in Eqs. 15 to 21) and on uncertainties in chlorophyll, non-algal particles, and CDOM concentrations. First-order autoregressive processes are used to generate 2D stochastic perturbation fields that are multiplied by the 2D fields of IOPs. The same perturbation is applied to all vertical levels. The method for perturbing the IOPs is detailed in Macé et al. (2025). The inputs for perturbations (standard deviations, temporal, and spatial correlations) must be provided in the `namsto` namelist. Given the perturbation using log-normal distributions, the mean $\mu_0$ and standard deviation $\sigma_0$ of the initial Gaussian distribution are given following Eqs. 22 and 23 with $\sigma$ the standard deviation of the final log-normal distribution centred on 1.

$$\mu_0 = -\frac{\sigma^2}{2} \tag{22}$$

$$\sigma_0^2 = ln(1 + \sigma^2) \tag{23}$$

## 3 Framework for application in the Black Sea

The RT module presented in Sect. 2, including its stochastic version, has been tested in the Black Sea where NEMO-BAMHBI is run in forecasting and reanalysis mode in the frame of the Copernicus Marine Service. Its inclusion as an additional module to the BAMHBI model, which upgrades the current version of BAMHBI, is presented in this section. The RT model has been calibrated thanks to Biogeochemical (BGC)-Argo floats that have been deployed in the Black Sea for more than a decade. The IOPs necessary to compute the spectral irradiance are provided based on NEMO-BAMHBI variables. The use of the RT module is presented in this section as an upgrade of the existing NEMO-BAMHBI system. We also introduce here the coupling of the RT module with the BAMHBI framework, such as the computation of IOPs and the surface radiative forcing.

### 3.1 The BAMHBI model

The BiogeochemicAl Model for Hypoxic and Benthic Influenced areas (BAMHBI) is a marine biogeochemical model that describes the cycles of carbon, nitrogen, phosphorus, silicon and oxygen through the modelling of the marine foodweb from bacteria up to mesozooplankton (Grégoire et al., 2008; Grégoire and Soetart, 2010). A complete technical description of the BAMHBI model can be found in Grégoire et al. (2025). It solves three PFTs, two zooplankton types and the microbial loop with several classes of detritic materials. The BAMHBI model is appropriate to model low-oxygen environments, explicitly representing the anoxic layer of the Black Sea. In the current version of BAMHBI, only a direct stream of light is described in three wavebands, including two in the visible part, differentiating radiation between a short and a long waveband in the visible range (Capet, 2014). The third band is in the infrared range. The absorption and scattering of light by the three phytoplankton groups and organic particles is considered as described in Grégoire et al. (2025) and the absorption of CDOM is parameterised as a function of salinity. We force the surface radiation from the regional configuration of an atmospheric model. At sea





surface, the reflected fraction of irradiance, due to surface albedo, is removed to obtain radiation just below sea surface. Below

the surface, a single irradiance stream $E_{fo}$ is attenuated following Beer's law with an absorption coefficient $a_{fo}$ derived from chlorophyll concentration, particulate organic carbon (POC), and salinity, in each of the three wavebands.

$$a_{fo} = a_{fo.w} + a_{fo.chl} + a_{fo.poc} + a_{fo.cdom} \tag{24}$$

$$\frac{dE_{fo}(z)}{dz} = -a_{fo}E_{fo}(z) \tag{25}$$

We aim to build on top of the current version of BAMHBI described in Grégoire et al. (2008) by extending it with the three-

stream RT scheme described in Sect. 2 as an additional optional module. The code for the extended version of BAMHBI is available in the supplementary material. When BAMHBI is used with the RT module, it provides the IOPs to solve the spectral direct and diffuse radiation according to Eqs. 1 to 6. Three PFTs are considered: dinoflagellates, smaller flagellates and diatoms. These species are dominant in the Black Sea (Silkin et al., 2021). Data for the reference spectra are adapted from Álvarez et al. (2022) to match the PFTs simulated in BAMHBI. Specific absorption and scattering spectra for phytoplankton are provided in

Fig. 1. $a_{phy}$, $b_{phy}$, and $b_{b,phy}$ are therefore computed as the sum of contributions from these three PFTs following Eq. 15 to 17. The optical properties of non-algal particles are computed using POC that is explicitly simulated in BAMHBI, following Eqs. 18 to 20.

CDOM is not explicitly simulated in BAMHBI but needs to be provided by a forcing function. This forcing is built from

using BGC-Argo data in the Black Sea between 2017 and 2020. Among the collected data, we use profiles of chlorophyll and particulate backscattering coefficient at 700 nm to derive the optical properties of phytoplankton and non-algal particles. We use CDOM profiles to impose a specific shape to the CDOM absorption profile, assuming that the absorption power of CDOM is uniform in the vertical dimension. Since BGC-Argo floats provide measurements of downward irradiance in three wavebands (centred on 380, 412 and 490 nm), we use the 1D test model to optimise CDOM absorption to best match the data. Given the

large amount of available BGC-Argo profiles, we maintain seasonality in the resulting CDOM absorption forcing. Rather than having a forcing depending on depth, we also set the dependence in density ($\rho$) to exhibit seasonal and spatial variations in the composition of seawater. Table 1 provides the values used in this NEMO-BAMHBI configuration.

When coupled, the RT model feedbacks to BAMHBI by providing the scalar photosynthetic active radiation (PAR). It is

defined as the scalar integrated irradiance in the visible range (i.e. between 400 and 700 nm) as in Eq. 26. PAR is the fraction of irradiance available to phytoplankton and is therefore defined from the three streams of irradiance. *In situ* observations of PAR are also often available, making it a useful variable for model validation.

$$PAR(z) = \int\limits_{400nm}^{700nm} \left[ \frac{E_d(\lambda,z)}{\overline{\mu_d}} + \frac{E_s(\lambda,z)}{\overline{\mu_s}} + \frac{E_u(\lambda,z)}{\overline{\mu_u}} \right] d\lambda \tag{26}$$



**Table 1.** Model parameters for the Black Sea implementation of the RT model.

| Parameter | Value | Unit | Source |
|---|---|---|---|
| $\overline{\mu_{d}}_{(max)}$ | 1.5 | [-] | Aas (1987); Dutkiewicz et al. (2015) |
| $\overline{\mu_{s}}$ | 0.83 | [-] | Aas (1987); Dutkiewicz et al. (2015) |
| $\overline{\mu_{u}}$ | 0.4 | [-] | Aas (1987); Dutkiewicz et al. (2015) |
| $r_{s}$ | 1.5 | [-] | Aas (1987); Dutkiewicz et al. (2015) |
| $r_{u}$ | 3 | [-] | Aas (1987); Dutkiewicz et al. (2015) |
| $Q$ | 4 | sr | Morel and Gentili (1993); Lee et al. (2002) |
| $T$ | 0.52 | [-] | Lee et al. (2002) |
| $\gamma$ | 0.425 | sr | Lee et al. (2002) |
| $S_{cdom}$ | 0.02 | 1/nm | Terzić et al. (2021) |
| $\eta_{H}$ | 0.85 | [-] | model calibration |

## 3.2 MAR surface forcing

The RT model needs information on the spectral irradiance at sea surface as a boundary condition. Before reaching the sea surface, the solar irradiance propagates through the atmosphere, where it is already divided into direct and scattered streams. This information is then propagated in the water column according to the IOPs. To provide these boundary conditions, we use spectral shortwave fluxes simulated over the Black Sea area with a regional atmospheric model (Gallée et al., 2013; Grailet et al., 2025). MAR (Modèle Atmosphérique Régional) is a regional climate model used for both weather forecasting and cli-

mate studies (Gallée et al., 2013). For this study, the atmosphere over the Black Sea has been simulated by the version 3.14 of MAR (Grailet et al., 2025), which runs with the ECMWF radiative transfer scheme ecRad v1.5.0 (Hogan and Bozzo, 2018). Operational since 2017, ecRad is a flexible radiation scheme which allows users to easily tune its sub-components, such as cloud and gas optics. In particular, ecRad v1.5.0 is capable of running with high-resolution gas optics schemes pre-computed with the ecCKD tool (Hogan and Matricardi, 2022). Since late 2022, it has also been capable of producing spectral shortwave

fluxes in user-defined bands. Grailet et al. (2025) demonstrated that running MAR v3.14 with ecRad and a high-resolution ecCKD gas optics scheme for the shortwave range can produce realistic spectral shortwave fluxes with respect to ground observations. The spectral bands can be as small as 5 nm wide, as long as they are not finer in resolution than the inner spectral bands of the ecCKD model.

To simulate the Black Sea, MAR has been configured with a 15 km grid resolution and 24 pressure levels in sigma coordinates that extend from the surface to the lower stratosphere. It has been forced at its lateral boundaries by ERA5 reanalyses (Hersbach et al., 2020) and has been tuned to use a high resolution ecCKD gas-optics scheme in the shortwave range, to produce fine spectral shortwave fluxes. The output spectral bands have been configured to be equivalent to the 33 wavebands used



in the ocean RT model, ranging between 250 and 4000 nm. For consistency, we also choose to force ocean physics of the model

with outputs from the same MAR simulation: sea level pressure, humidity, 2 m temperature, wind speed and precipitation.

It should be noted that any potential bias in the forcing would not significantly influence the fields of sea surface reflectance that are produced. In fact, reflectance is computed as the ratio of upwelling irradiance to downwelling irradiance (see Eq. 7). This equation can also be understood as the normalisation of upwelling irradiance, influenced by absorption and backscatter-

ing, by the forced downwelling irradiance. Despite a difference in incident downwelling irradiance, the fraction of irradiance that is backscattered remains the same, and therefore the ratio between upwelling and downwelling irradiance at sea surface would remain rather similar.

Radiative forcing consists of downward direct $E_{d.forc}$ and scattered $E_{s.forc}$ irradiance streams on the sea surface for each

wavelength and grid location of the domain. We derive the boundary conditions described in Eqs. 4 and 5 by accounting for sea surface albedo $A$, taken from the mean monthly albedo dataset for the Atlantic Ocean at 40°N described in Payne (1972).

$$E_{d0} = (1 - A)E_{d.forc} \qquad (27)$$
$$E_{s0} = (1 - A)E_{s.forc} \qquad (28)$$

### 3.3 Description of the use case configuration

Based on the initial scheme with simple optics and the three-stream RT scheme, we define three BAMHBI configurations that are described here and analysed in section 4:

- Simple optics: using the one-stream scheme from Capet (2014), detailed in section 3.1. The simple optics scheme is the one that runs with the current version of BAMHBI (Grégoire et al., 2025).

- RT: using the updated "radtrans" model from Dutkiewicz et al. (2015), a three-stream model in thirty-three wavebands,
as described in section 2, coupled with NEMO-BAMHBI.

- Stochastic RT: using the stochastic mode of the RT model, presented in section 2.4, and therefore accounting for uncertainties in the modelling of IOPs. The corresponding experiments consist of a 50-member ensemble used to compare time series of sea surface reflectance with *in situ* data.

With the inclusion of the three-stream RT scheme and the new surface radiative forcing, the deterministic and stochastic

RT configurations presented in this article make an extension of the NEMO-BAMHBI modelling framework. The RT module takes inputs from BAMHBI and external forcings, and its outputs are used by both NEMO and BAMHBI. It includes the computation of three new state variables that correspond to the three streams of irradiance, and a diagnostic variable that is the sea surface reflectance.





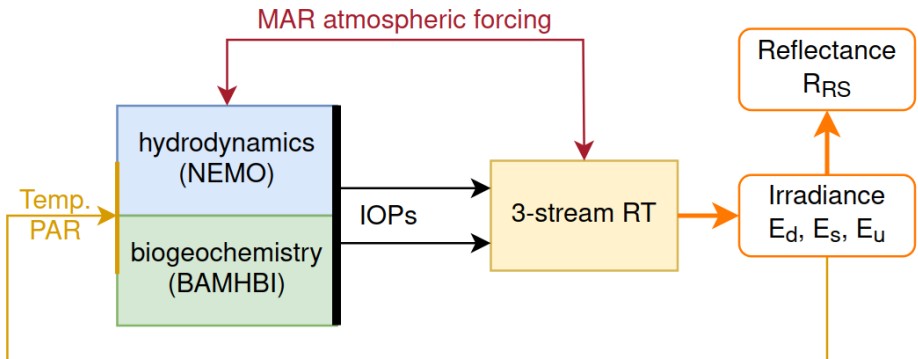

**Figure 2.** Coupling of the 3-stream RT model within the NEMO-BAMHBI framework. Forcing for physics and surface radiation are provided by a regional MAR simulation (see Section 3.3). In a configuration without a biogeochemical model, the IOPs would be forced from external data into the three-stream RT module

In all experiments, BAMHBI is coupled with NEMO 4.2 on the same horizontal and vertical grids. We use a NEMO configuration for the Black Sea with a horizontal resolution of 15 km and 59 vertical levels distributed unevenly. The layers are thinner close to the surface and wider in the deeper parts of the basin. Biogeochemical forcings such as river inputs and atmospheric deposition of nutrients (phosphorus, nitrate and ammonium) are based on climatology data. The model is forced with river runoffs that are based on climatology. We assume that there are no exchanges with the Sea of Azov, and the exchanges with the Sea of Marmara at the Bosphorus Strait are set following Stanev and Beckers (1999). In the RT configurations, we consider 33 wavebands ranging between 250 and 4000 nm with a finer 25 nm resolution in the visible range. Temperature and PAR feedbacks are activated with NEMO and BAMHBI, respectively. In this configuration, the parameter $\eta_H$ (see Eq. 10) transcribing the fraction of absorbed irradiance that is used to heat the water column is set to 0.85 after calibration of the model with a 1D test model in the Black Sea. Figure 2 provides an overview of the coupled modelling framework for experiments using the RT module.

The parameters for the RT module specific to the use case are specified in the `nam_RADTRANS` namelist. They consist of the parameters defined in Table 1 along with the forced inputs:

– The wavebands and spectral range to be simulated

– The parameters described in Sect. 2: $r_s, r_u, \mu_d, \mu_s, \mu_u, Q, T, \gamma, \eta_H$

– The absorption and scattering spectra for water, phytoplankton and non-algal particles

– The reference absorption profile for CDOM absorption and the slope coefficient $S_{cdom}$

– The surface radiative forcing (downward direct and scattered streams)





## 4 RT modelling in the Black Sea

This section describes the application of the RT model to the Black Sea through the NEMO-BAMHBI framework. We first investigate the consequences of substituting the RT scheme in the computation of temperature and chlorophyll. This is done through comparison with BGC-Argo and satellite data, and with the simple optics configurations when available. We then elaborate on the computation of radiometric variables, with an outlook on the uncertainties that arise from the parameterisation of the IOPs. We highlight the benefit of a RT model for linking simulations with observations with the example of a chlorophyll estimate derived from the RT simulated reflectance as estimated in ocean colour inversion algorithms.

### 4.1 In-water irradiance

In the simple optics configuration, PAR is simulated from two wavebands in the visible range and with a simple one-stream irradiance model. With the RT configuration, we simulate the irradiance at a higher spectral resolution, providing a more detailed representation of the spectral irradiance. Scatter plots of downwelling irradiance at 380, 412 and 490 nm simulated by the RT model and observed by BGC-ARGO (between 2017 and 2020) are presented in Fig. 3. In the model, the downwelling irradiance is defined as the sum of the direct and downward scattered streams of irradiance. The data shown are the logarithms of irradiances in order to compare orders of magnitude, as a way to account for both higher values at the surface and lower values in the depth. We find that the simulation of the irradiance streams is very consistent with the observed data despite an underestimation of irradiance at 380 nm. The agreement in the 412 and 490 nm wavebands is very good with regression slopes close to 1. The bias is larger at 380 nm and decreases with the wavelength while RMSEs are of similar magnitude.

It should be noted that the method used to compute PAR is not the same in both configurations, leading to differences at the sea surface even though the forcing is the same. In the simple optics configuration, PAR is defined as 46 % of the total solar radiation, while in the three-stream model it is defined as the integration of solar radiation between 400 and 700 nm. The surface PAR tends to be higher in the RT configuration. To analyse the attenuation of PAR profiles, we focus here on normalised profiles with regard to the surface values. Figure 4 presents a scatter plot with data from normalised PAR profiles simulated by the simple optics and the RT configurations with BGC-Argo data between 2017 and 2020. The agreement with BGC-Argo data for normalised PAR is very good, with a regression slope in the scatter plot of over 0.9. Since this slope is lower than 1, the difference between model outputs and BGC-Argo data is explained by a higher absorption in the model close to the surface and too low in depth. The use of a RT model slightly improves the simulation of the PAR profile with lower error and bias. The performance of both models remains rather close and the change in scheme substitution does not significantly influence the vertical profiles of PAR.





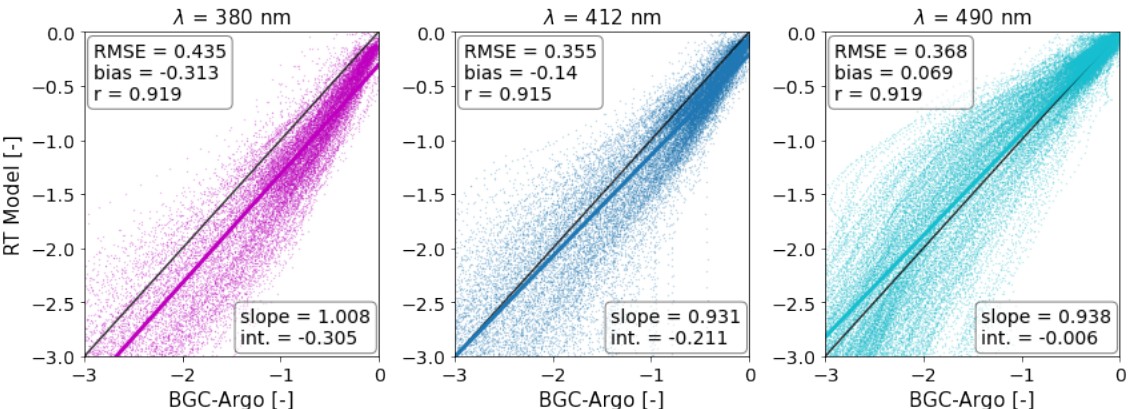

**Figure 3.** Logarithm of normalised downwelling irradiance from BGC-Argo data and the RT model in the RT configuration at the location of BGC-Argo floats between 2017 and 2020, for the upper 100 metres. Irradiance measurements are normalised by the closest available value to the surface. The indexes of the BGC-Argo floats used are 6901866, 6903240 and 7900591. RMSE, correlation coefficient (r), regression slope and intercept are displayed for each wavelength. Scatter plots show 19 556, 25 628 and 43 268 data points respectively for the 380, 412 and 490 nm wavebands.

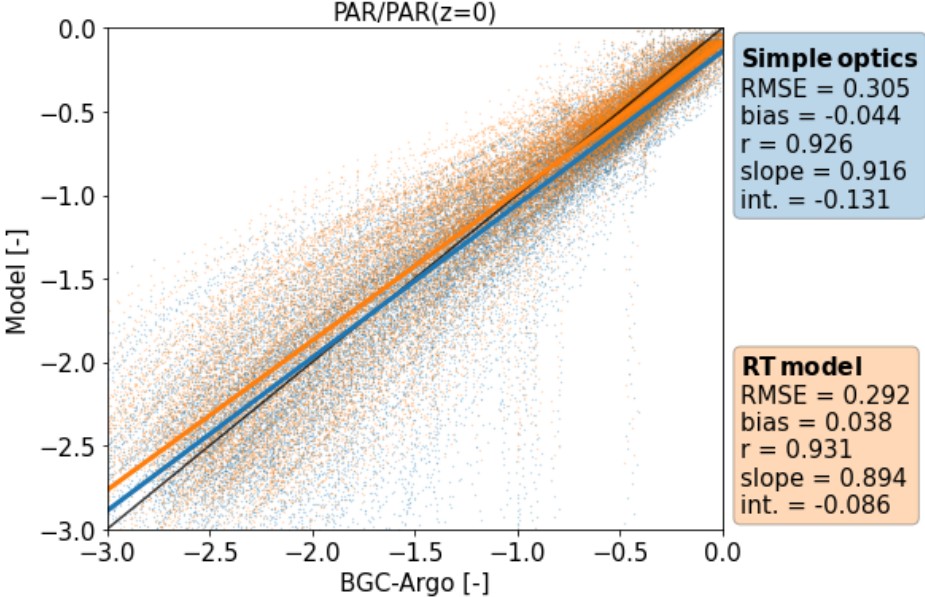

**Figure 4.** Logarithm of normalised PAR from BGC-Argo data with the simple optics and RT configurations at the location of BGC-Argo floats between 2017 and 2020, for the upper 100 metres. RMSE, bias correlation coefficient (r), regression slope and intercept are displayed. Scatter plots show 36 376 data points. Normalisation is performed relative to the maximum value of the PAR profile.





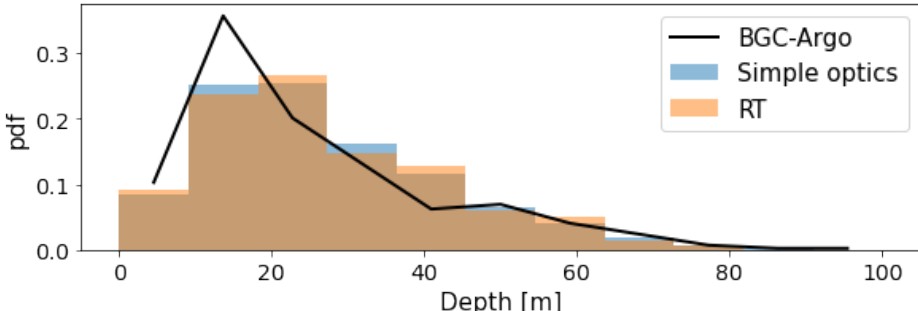

**Figure 5.** Distributions of thermocline depth compared with data from BGC-Argo floats between 2017 and 2020 in the simple optics and RT configurations, respectively in blue and orange. Bins are taken following the vertical grid of the model. 418 profiles are considered for thermocline depth.

## 4.2 Simulation of temperature profiles

In this section, we assess the influence on temperature of solving radiative transfer with a three-stream RT scheme that solves the direct and diffuse (upward and downward) radiation in a large number of spectral bands. In the coupling with NEMO, the irradiance streams are used to compute the evolution of temperature in the model following Eq. 10. This implies that a change in the irradiance streams influences physics, which in turn influences the biogeochemistry and IOPs. Calibration of the $\eta_H$ parameter, which intervenes in Eq. 10, is performed in such a way that the temperature profiles in the RT configuration remains

consistent with BGC-Argo profiles for our Black Sea configuration.

To evaluate the influence of this feedback on temperature from the RT to the physics, we consider the depth of thermocline, defined as the depth of the greatest temperature gradient in a temperature profile. The determination of the thermocline in the model is constrained by its vertical resolution, which means that the modelled thermocline depth can only be taken among a

set of discrete values. We use outputs from the simple optics and RT configurations, that are compared with BGC-Argo data. Figure 5 compares the distributions of thermocline depth as estimated in the simple optics and RT configurations, and measured by Argo floats. Bias, root mean square error and correlation statistics are presented in Table 2.

Using a spectral RT model to compute the source of energy in the temperature equation does not significantly influence the

distribution of the thermocline depths and slightly improves the agreement with *in situ* data. It should be noted that the simple optics configuration has already been tested and well calibrated, and no significant improvement is expected in this regard by changing the RT scheme. A positive bias remains, exhibiting that the simulated thermocline is too deep by about 1.40m. This bias is acceptable considering the vertical resolution of the model which is about 5 m at the depth of the thermocline. Similarly, no significant changes between the two configurations are observed in the simulation of the cold intermediate layer

cold content (CCC) and the mixed layer depth over the four years of simulation.





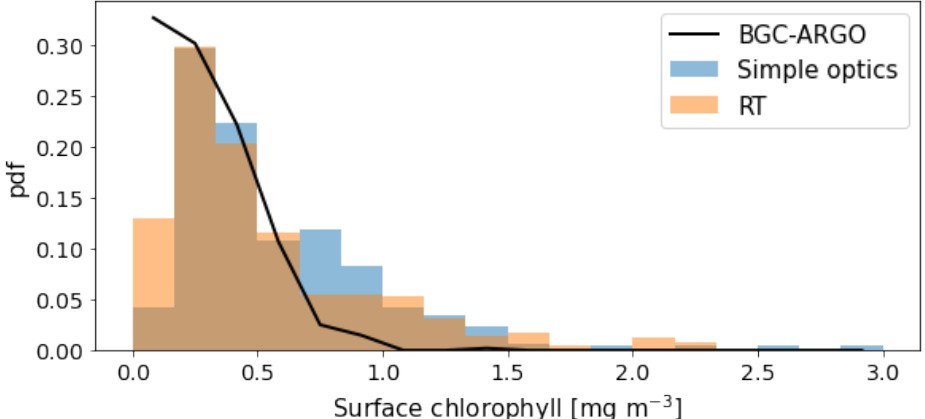

**Figure 6.** Distributions of surface chlorophyll compared with data from BGC-Argo floats between 2017 and 2020. 478 profiles are represented here for the simple optics and RT configurations.

### 4.3 Simulation of chlorophyll

As for temperature, the IOPs of the upper ocean layers dictate the amount of light received by the lower layers, and therefore the ability for phytoplankton to develop in the deeper parts. Figure 6 compares the distributions of surface chlorophyll simulated in the RT configuration and from BGC-Argo data. For comparison, results from the simple optics configuration are given. The

BGC-Argo chlorophyll dataset is corrected following Ricour et al. (2021), to reduce bias in chlorophyll profiles. The statistics for this comparison are presented in Table 2.

The use of the RT module slightly improves the simulation of surface chlorophyll, although some bias remains. The series of surface chlorophyll is better correlated with the measured data, and the RMSE decreases by 18%. As shown in Fig. 6, the

model still tends to overestimate surface chlorophyll concentration on average with a positive bias that remains. One of the reasons for this bias is the overestimation of blooms in the biogeochemical model in winter and spring, which is not corrected by using the RT module.

### 4.4 Sea surface reflectance

The most important addition that comes with the RT configuration is the simulation of sea surface reflectance fields. We use remote-sensing data at the basin scale to assess the ability of the model to reproduce fields of sea surface reflectance. Figures 7, 8, and 9 respectively present the monthly distributions of $R_{RS}$ at $\lambda = 490$, 555 and 670 nm in simulated and remote-sensing data, in 2018. Data are taken from the daily satellite product throughout the basin. In the beginning of the year, we notice that the distribution of reflectances at 490 nm is more spread out in the satellite data than in the simulation. The model is able, on



**Table 2.** Bias, RMSE and correlation of simulated variables compared with BGC-ARGO data. Profiles are taken from floats 6901866, 6903240 and 7900591 between 2017 and 2020.

| Variable | # of profiles | Model | Bias | RMSE | Corr. |
|---|---|---|---|---|---|
| Thermocline depth | 418 | Simple optics | 1.52 m | 12.39 m | 0.70 |
| | | RT | 1.40 m | 12.25 m | 0.71 |
| Normalised PAR | 478 | Simple optics | -0.015 | 0.059 | 0.96 |
| | | RT | -0.004 | 0.060 | 0.96 |
| Surface chlorophyll | 221 | Simple optics | 0.32 mg m$^{-3}$ | 0.31 mg m$^{-3}$ | 0.26 |
| | | RT | 0.25 mg m$^{-3}$ | 0.50 mg m$^{-3}$ | 0.29 |

average, to simulate the correct $R_{RS}(490)$, but is not always able to reach higher and lower values for reflectance. Noticeably in February, some high values of reflectance are missing. They correspond to a localised bloom on the northeastern coast of the basin that is underestimated in the model. From May to July, the model largely underestimates $R_{RS}(490)$. This corresponds to the period during which coccolithophores bloom in the Black Sea (Kubryakov et al., 2021). This signal is not picked up by the model, which explains the bias for those three months. From August and until the end of the year, distributions from the model are mostly in agreement with the data despite a noticeable overestimation of sea surface reflectance, hinting at the fact that the model is more reliable when simulating reflectance outside of blooming conditions.

At 555 nm (Fig. 8), we also notice a thinner spread in the simulated reflectances compared to the satellite data at the beginning of the year. The March and April panels seem to indicate that a bloom is observed later in the model than in the observations. The influence of the coccolithophore bloom that is not picked up by the model is lesser at 555 nm, and insignificant at 670 nm. Then, we find distributions of reflectances that are consistent with observations from August and until the end of the year. At 670 nm, we notice lower reflectances in particular because of higher absorption by water. The distributions are consistent for most of the year until some differences appear in the autumn. At this wavelength, the influence of phytoplankton and CDOM is much lower and non-algal particles are the main driver of optical properties. The overestimation of reflectances in Autumn could indicate concentrations of non-algal particles that are higher in the model than observed, thus leading to increased backscattering and reflectance signal.

In general, the model is able to simulate the main trends in sea surface reflectance at the basin scale. The agreement is best at longer wavelengths such as 670 nm, where the signature of phytoplankton is less dominant. At 490 and 555 nm, some features are misrepresented. The main differences come from the intensity of the blooms and from delays. The model presents less variability compared to the satellite product, keeping the simulated reflectances close to their mean seasonal values. The 490 nm wavelength falls within the spectral range of high backscattering by phytoplankton, allowing us to assess the influence of blooms on the simulated reflectance here. The 555 nm wavelength also provides valuable information on blooms in





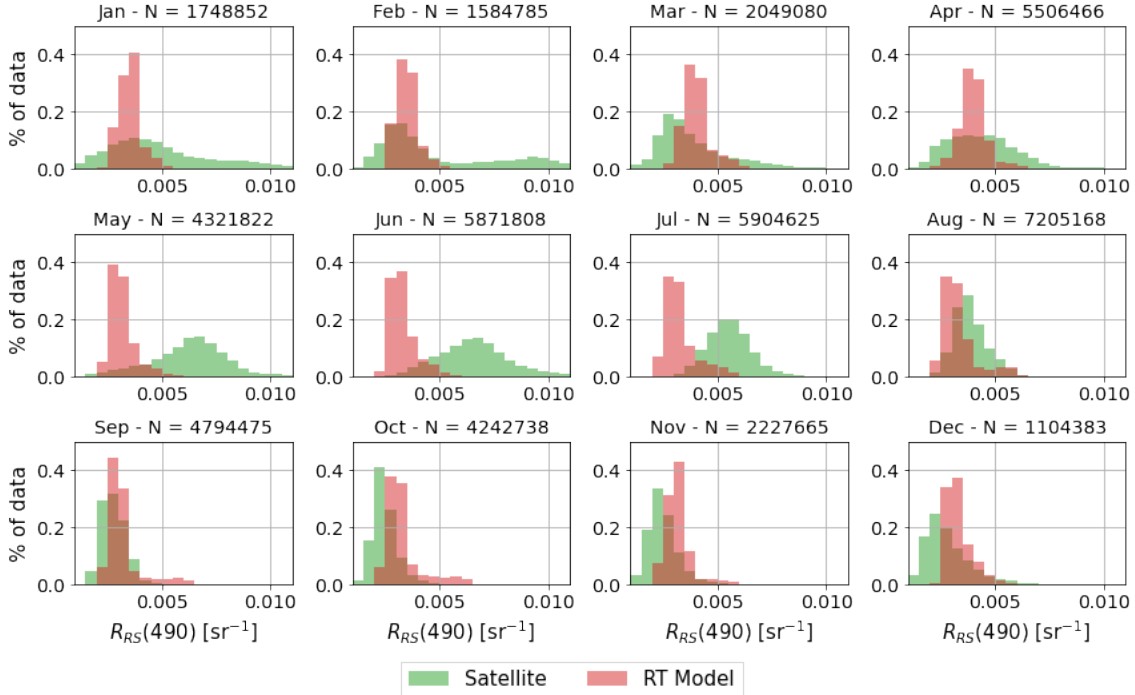

**Figure 7.** Monthly distributions of sea surface reflectance at 490 nm in 2018 across the Black Sea basin. Simulated reflectance is interpolated at the location of available remote-sensing data.

the basin, with also a lower influence of CDOM than at 490 nm. In the 670 nm waveband, the agreement between simulated
reflectances and observations appears to be better throughout the year.

We compare for the whole basin in Fig. 10 that shows maps of $R_{RS}$ at 490 and 555 nm for the 27th October 2018 on the
left and central panels. This date is chosen because of the absence of clouds that limits the spatial coverage of the satellite
product, and to illustrate a situation where we observe bias in the reflectances, but not significantly in the reflectance-derived
chlorophyll. As expected, we notice higher reflectances in the northwestern shelf where the biological activity is higher for
most of the year. In late October, outside of blooming conditions in the deep basin, we notice a rather low bias when comparing
with remote-sensing data. This bias is here positive, which is in agreement with the pattern evidenced in Figs. 7 and 8. The
bias is higher on the shelf where the increase in reflectance in the model (compared to the deeper areas of the basin) is likely
too high compared to observations. It should be noted that the bias on the shelf is rather high, of the same order of magnitude
as the absolute values of reflectances.





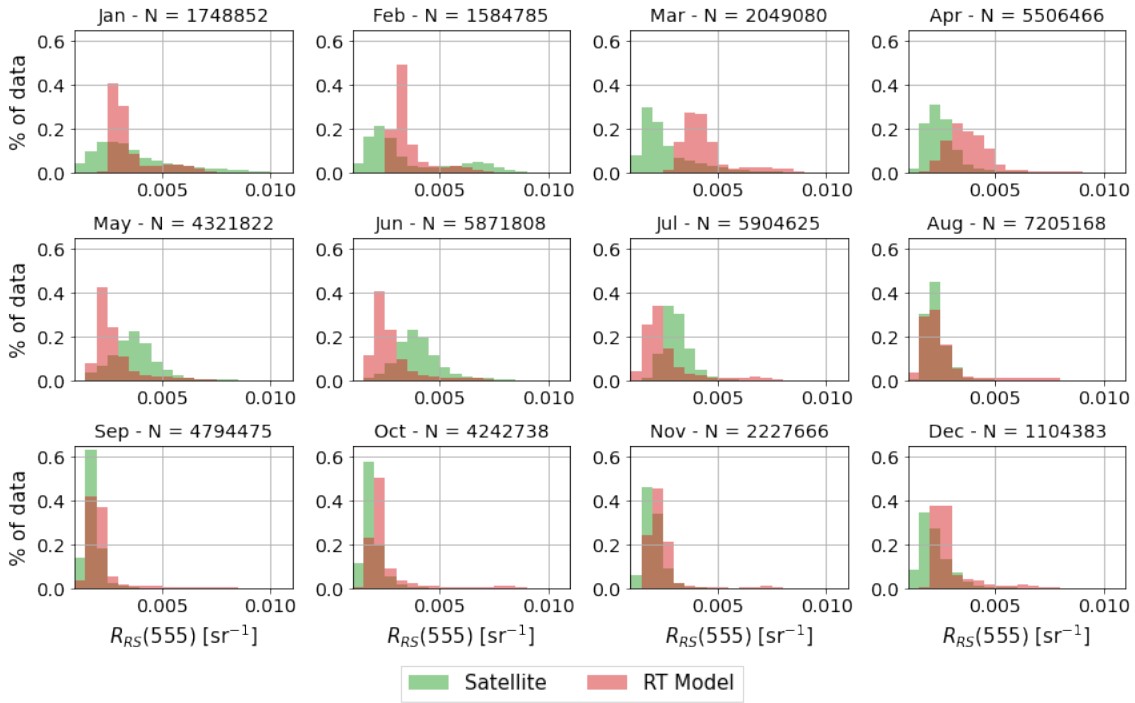

**Figure 8.** Monthly distributions of sea surface reflectance at 555 nm in 2018 across the Black Sea basin. Simulated reflectance is interpolated at the location of available remote-sensing data.

## 4.5 Reflectance-derived surface chlorophyll

Algorithms have been developed to derive biological quantities from reflectance fields, in particular to take advantage of remote-sensing data. In the Black Sea, Zibordi et al. (2015) proposed a method to provide surface chlorophyll concentration

fields based on reflectance. This method combines a band-ratio algorithm and a neural network approach that is mainly used for more complex coastal waters. This combined method is used by the Copernicus Marine Service to provide surface chlorophyll products for the Black Sea. Using the reflectance simulated by our RT model, we can mimic the band-ratio algorithm in order to compute a new estimate of surface chlorophyll concentration. This algorithm uses the 490 and 555 nm wavelengths, respectively representative of blue and green (Kajiyama et al., 2018). In the following, we refer to this surface chlorophyll

estimate as reflectance-derived chlorophyll rCHL:

$$log(rCHL) = \sum_{k=0}^{3} c_k \times \left[ log\left( \frac{R_{RS}(490)}{R_{RS}(555)} \right) \right]^k \tag{29}$$

The coefficients $c_k$ are provided in Kajiyama et al. (2018) for the Western Black Sea. We extrapolate and use these coefficients for the entire basin here. Reflectance-derived chlorophyll is not independent of the chlorophyll dynamically simulated in





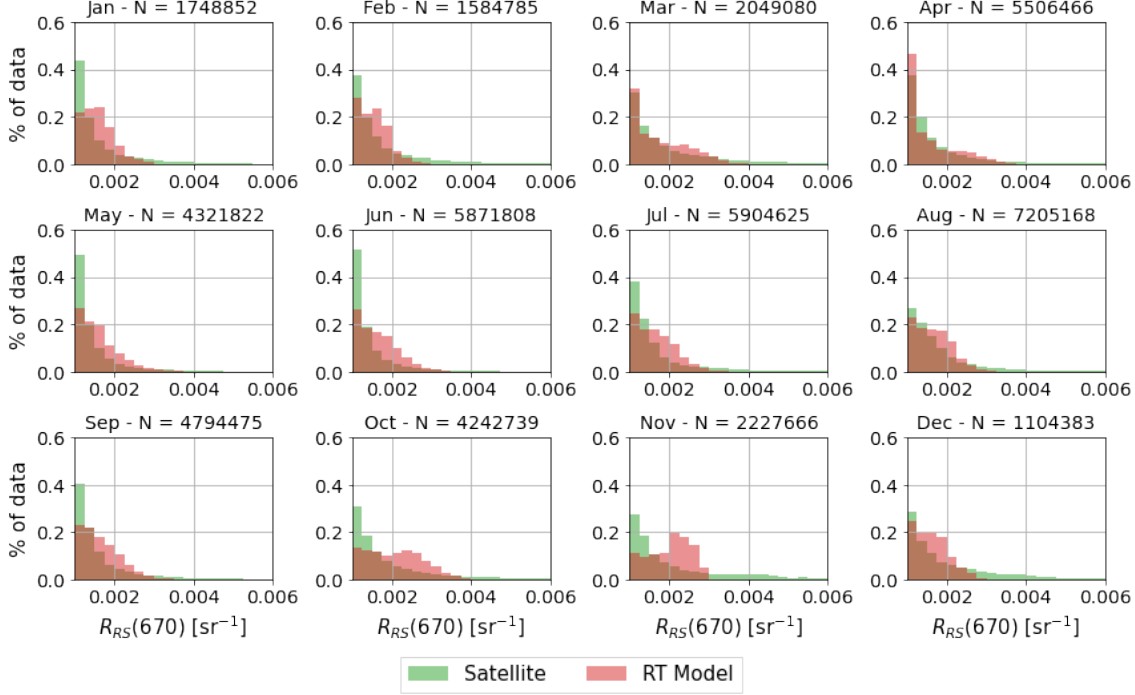

**Figure 9.** Monthly distributions of sea surface reflectance at 670 nm in 2018 across the Black Sea basin. Simulated reflectance is interpolated at the location of available remote-sensing data.

BAMHBI because the latter intervenes in the computation of IOPs. However, it provides a quantity that is more closely linked
to satellite data by its very definition, using reflectance data. We note that using sea surface reflectance ratios also reduces the uncertainty in the final variable, since the approximation made on the BRDF coefficient $Q$ is removed.

The right-hand side panels of Fig. 10 show the resulting field of $rCHL$ for the 27th of October 2018 and the deviation with the satellite product. In this case, our model tends to underestimate the surface chlorophyll concentration in the deep basin and
overestimate it in coastal areas. Although the coastal overestimation is rather consistent seasonally, the underestimation in the deep basin occurs primarily during autumn and winter, whereas a slight overestimation tends to occur in spring and summer, in agreement with the distributions of Fig. 11.

When comparing the distributions of $rCHL$ from our simulation and from the satellite product in Fig. 11, we notice pat-
terns that are very similar to those observed with sea surface reflectance at 490 nm. In this figure, we display both $rCHL$ and the chlorophyll computed dynamically in BAMHBI for reference, considering the mean optical depth of the Black Sea of 10 metres (Peneva and Stips, 2005). Chlorophyll concentration can first represent blooms in winter, but then overestimates the magnitude. The estimate of surface chlorophyll remains higher than observations during summer until it agrees well with



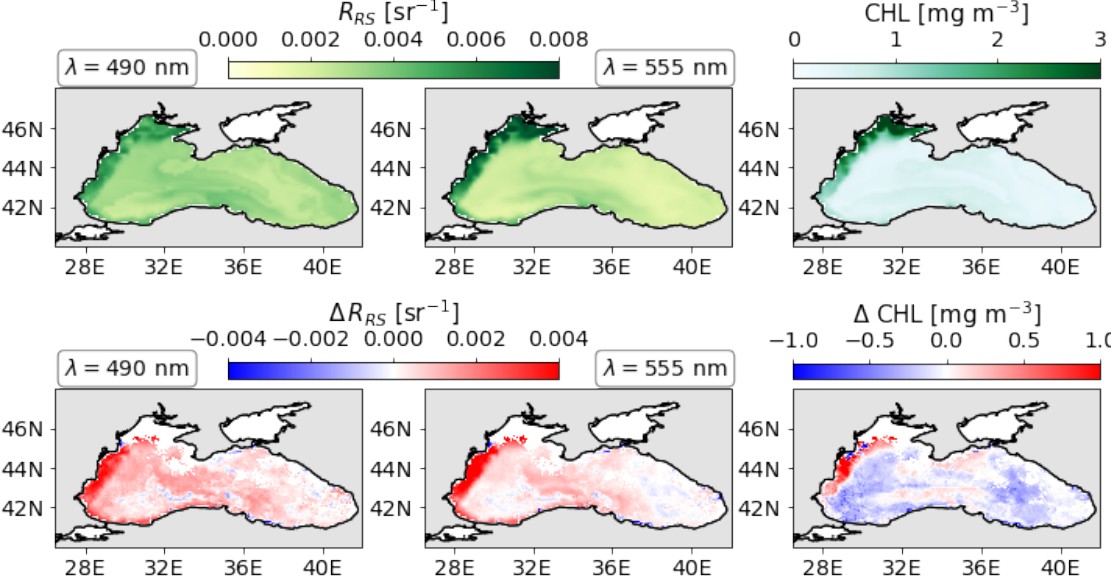

**Figure 10.** On the top row, maps of simulated $R_{RS}(490)$ (left), $R_{RS}(555)$ (centre) and rCHL (right) for the 27th October 2018. On the bottom row, difference with the remote-sensing product.

data between September and the end of the year. We notice that the distributions of $rCHL$ are very different from those of
the BAMHBI chlorophyll, which is often too high compared to the remote-sensing data. $rCHL$ agrees better with the satellite
product than BAMHBI chlorophyll for most of the year. It should also be noted that, surprisingly, the coccolithophore bloom
that is identified in the satellite product of sea surface reflectance does not appear in the surface chlorophyll signal. With the
band-ratio algorithm, increases in reflectance in both wavebands cancel out, providing low concentrations. In such conditions,
we could assume that the satellite product may be biased.


    Figure 12 shows the seasonal evolution of the RMSE between the satellite chlorophyll and the model estimated chlorophyll
in the simple optics and RT configurations, along with the reflectance-derived chlorophyll $rCHL$ from the RT configuration.
and satellite data are presented in Fig. 12 for the estimate of surface chlorophyll. The correlations with the satellite product
for all three series are very similar, close to 0.45. Although a change in the RT scheme does not significantly influence sur-
face chlorophyll as computed by BAMHBI, such as illustrated in Fig. 6, we notice a large drop in RMSE for $rCHL$. The
improvement is particularly important during the early spring bloom and is also significant during the rest of the year. Finally,
the standard deviation in the surface chlorophyll datasets is much higher than with $rCHL$. This seems to indicate that $rCHL$
does not tend to overestimate or underestimate surface chlorophyll as much as BAMHBI chlorophyll. On average, it produces
better estimates and thus leads to a decreased error throughout the year.



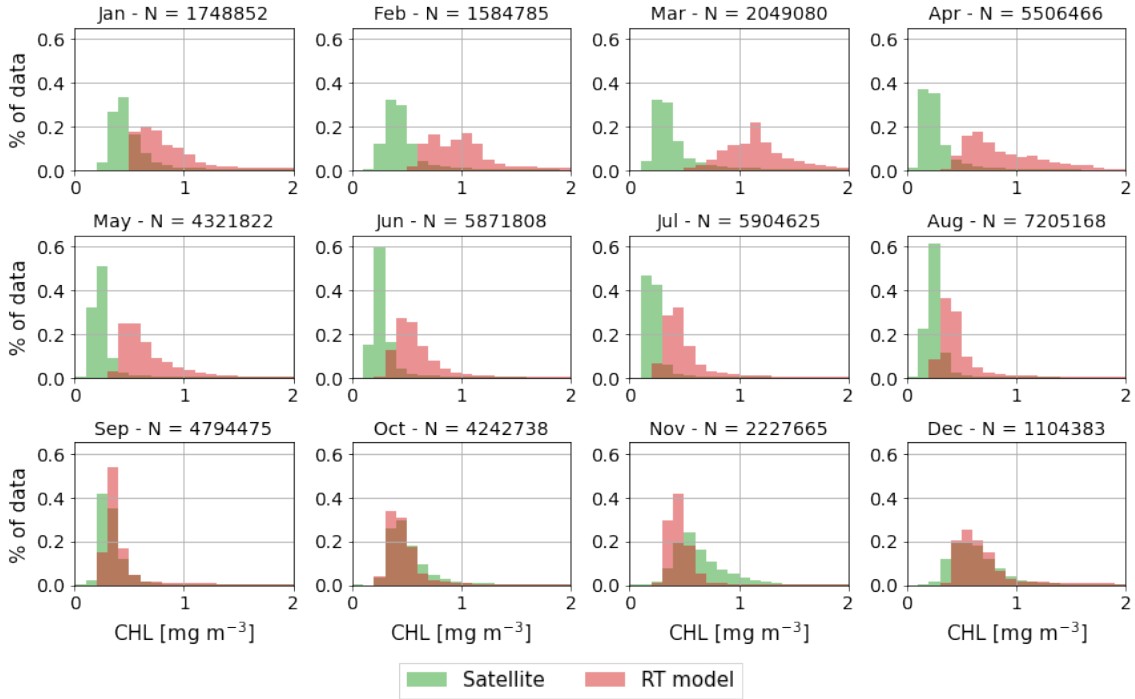

**Figure 11.** Monthly distributions of reflectance-derived chlorophyll in 2018 across the Black Sea basin. Simulated rCHL from the RT configuration is interpolated at the location of available remote-sensing data.

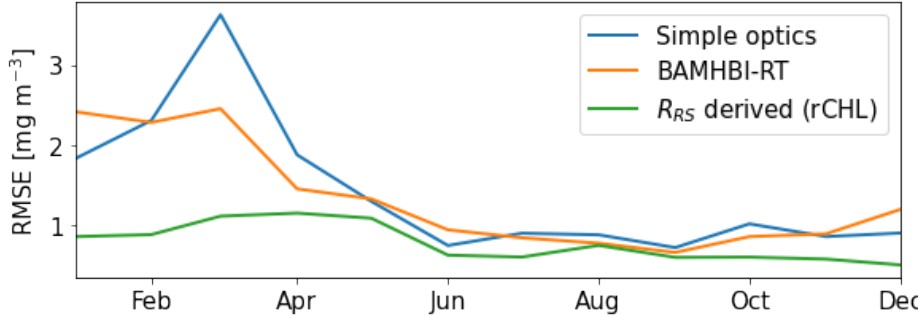

**Figure 12.** RMSE for chlorophyll over 2018 across the whole basin. Chlorophyll from the simple optics and RT configurations are presented. Reflectance-derived chlorophyll ($R_{RS}$) is computed following the inversion algorithm used to produce for satellite data in the Black Sea.

**4.6 Stochastic version of the RT model**

In the stochastic version of the RT model, the optical properties of phytoplankton, non-algal particles and CDOM are perturbed following the experiments described in Macé et al. (2025). We perturb IOPs with first order autoregressive processes with a time correlation of one month and a space correlation of approximately 75 km. The perturbations are defined with a standard





deviation of 30% for phytoplankton and non-algal particles, and 50% for CDOM. For each of the 50 members, we average the
reflectances for each month in order to create composites.

We compare the outputs of the stochastic configuration of the model with observations from the Galata and Gloria observation towers from the AERONET-OC network (Zibordi et al., 2006, 2009). They provide *in situ* measurements of water-leaving irradiance close to the western coast of the Black Sea, from which sea surface reflectance is derived. We use data from the
Galata platform to compare our reflectance fields over the year 2018 as in the reflectance spectra presented in Fig. 13. In addition to the *in situ* measurements performed at the Galata station, we also compare it to remote-sensing reflectance provided by the Copernicus Marine Service at the location of the Galata station. We also take advantage of the large dataset provided by this station to explore how the introduction of uncertainties in the RT model is useful when it comes to comparing sea surface reflectance, using the stochastic RT configuration. Figure 13 shows the monthly distribution of the surface spectral reflectance
simulated by the stochastic RT module and observed by satellite and at the Galata station. The standard deviation of the ensemble and the extreme values are shown. For observations, values are averaged in each monthly composite and the variability of reflectance within a month is shown using an error bar of one standard deviation length around the average monthly values.

In the observations, we notice few differences between the average *in situ* and remote-sensing data, but rather in the extrema
reached, indicating a wider distribution of the data in the remote-sensing product. Therefore, both datasets are rather consistent. The model is able to reproduce the main patterns of $R_{RS}$ spectra for 2018 in agreement with both sources of data, although some seasonal bias remains. In 2018, *in situ* data show two early blooms in January and March that are both picked up by the model. However, the simulated reflectance remains high in February between the blooms, indicating a potential merging of these blooms in the model, that is unable to separate them. The extrema members of the ensemble are able to get close to
the observations, but the ensemble mean does not fully capture the extent of the variability induced by the blooms. In spring and summer, the simulated reflectance is in agreement with the data, with observations falling within one standard deviation of the ensemble mean. In autumn, the ensemble tends to overestimate $R_{RS}$ with observations falling at the limit of the ensemble spread, but outside of one standard deviation. The agreement becomes good again at the end of the year with observations close to the ensemble mean in December. The model also represents the gradual increase in reflectance from October to December.
The maximum of reflectance is observed around 550 nm, in agreement with both datasets.

## 5   Discussion

In this paper, we present a spectral RT model that simulates the propagation of irradiance along the upward and downward vertical directions in three streams. The main outputs of the RT model are the spectral irradiances and the sea surface reflectance, which are quantities measured by radiometric sensors onboard satellite, BGC-Argo floats, and coastal stations. It links simu-
lated variables and observations of sea surface reflectance, avoiding the use of uncertain inversion algorithms. The RT model is integrated into the NEMO hydrodynamical model, where it is used in the computation of temperature as the heat source in the



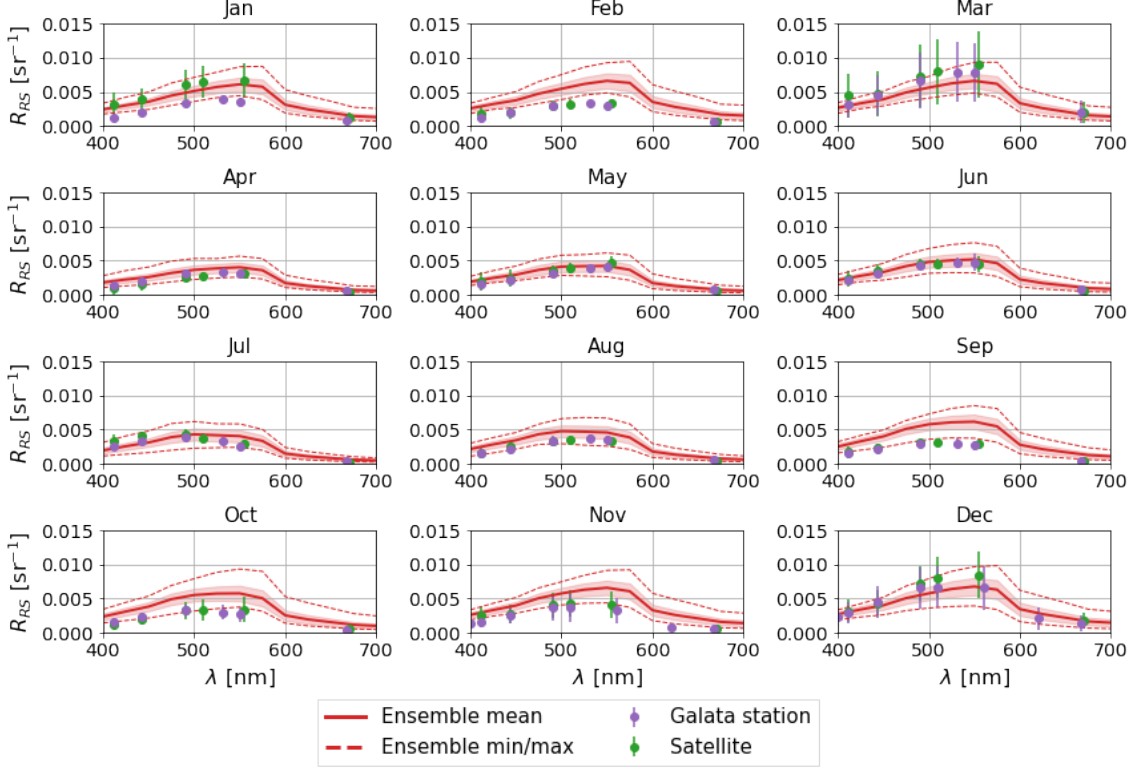

**Figure 13.** Monthly composites of sea surface reflectance spectra in 2018 at the Galata station (43.045°N, 28.193°E). Remote-sensing data are taken at the closest grid point to the Galata station. For observations, points represent the average $R_{RS}$ for the month and bars represent one standard deviation in the data of the month. Model results are taken from the stochastic RT configuration with the ensemble mean in bold lines, the ensemble standard deviation in shaded, and the ensemble minimum and maximum in dotted lines.

energy conservation equation is taken from the simulated irradiance streams. The vertical propagation of the spectral irradiance is governed by the water IOPs that determine the amount of light that is absorbed and scattered in the forward and backward directions. The water IOPs need to be provided to the RT model, either from a coupled biogeochemical model or from external datasets. A stochastic version of this RT model that accounts for the uncertainties in the IOPs is also provided.


The RT model and its stochastic version are tested in the Black Sea where they are coupled with NEMO and the biogeochemical model BAMHBI. The quality of the simulated radiometric variables, along with temperature and chlorophyll is assessed with comparison with satellite and BGC-Argo data. The modelling of in-water irradiance, PAR and temperature are consistent with observations, showing low to moderate bias and a high correlation as demonstrated in Figs. 3, 4 and 5. The quality of the simulated chlorophyll is lower but still acceptable, as it stays consistent with other configurations of the BAMHBI model in the Black Sea. We note a slight improvement in the simulation of PAR and thermocline depth when using the RT model compared to the configuration using a simple optics radiative scheme. The mean error in the representation of chlorophyll is also slightly






lower over the basin, as represented in Fig. 12.


The RT model explicitly simulates spectral radiometric quantities, enriching the modelling capabilities of the coupled model by providing spectral irradiance and sea surface reflectance being an important one. This is a key milestone towards comparisons between simulation outputs and remotely-sensing products. In common practice, comparisons are performed using remote-sensing surface chlorophyll products that require the use of inversion algorithms (e.g. Kajiyama et al., 2018). Those

algorithms tend to come with uncertainties, as they extrapolate from limited data onto larger basins. The simulation of sea surface reflectance is a first step toward the direct simulation of what is observed by satellites, thus removing the need for inversion algorithms. Figures 7, 8, and 9 show that the model is able to capture the main seasonal patterns of sea surface reflectance, but still have localised errors, in particular in blooming conditions. At longer wavelengths such as 670 nm (Fig. 9), where water and non-algal particles dominate the optical properties, the simulated distributions of irradiance are more consistent with ob-

servations. The simulation of reflectances also allows us to mimic inversion algorithms by using them on the simulated fields. As such, it provides a relevant framework for estimating uncertainties associated with surface chlorophyll, both in the model and in the satellite products.

In particular, the modelling of sea surface reflectance opens the way towards the direct assimilation of radiometric data. In

general, surface chlorophyll is the satellite product assimilated in biogeochemical models (e.g. Santana-Falcón et al., 2020), with *in situ* data such as BGC-Argo profiles also assimilated in studies (e.g. Teruzzi et al., 2021). There have been first attempts to assimilate reflectance data (e.g. Jones et al., 2016) that show promising results. The stochastic version of the RT model is particularly relevant in this context as it provides a tool to estimate model uncertainty, which is critical information for data assimilation. The estimation of uncertainties in the outputs of the RT model has already been discussed in (Macé et al.,

2025), where ensemble simulations were run to evaluate the consequences of the introduction of uncertainties in the IOPs. Other sources of uncertainty in the fully coupled modelling framework would have to be evaluated, such as the many empirical parameters used in biogeochemistry or the intrinsic model variability.

The use of the RT model significantly increases the computation time within the NEMO framework. RT is computed indi-

vidually at each time step, each waveband and for each ocean water column in the model. We find that the computation time is approximately doubled with 33 wavebands compared to the simple optics configuration. While this may not be an issue for short runs that span a couple of years in regional configurations, the inclusion of this RT model becomes costly for long-term simulations, global or high-resolution runs, in particular if ensembles are simulated. The computation time could be reduced by reducing the spectral resolution or by only considering backscattering and not forward scattering, thus simplifying Eqs. 1 to

3. In the configuration used for the test case, it is necessary to run the RT model at each time step because of the connections to hydrodynamics and biogeochemistry in the computation of temperature and PAR. However, users interested in simulating sea surface reflectance with a more simple one-way coupled configuration could run it less often to provide the desired outputs. Despite the increased computation time, the use of this system is relevant for modelling spectral irradiance and reflectance in





specific wavebands to focus on water constituents. Then, the use of the stochastic version seems particularly relevant in the context of data assimilation or to test parameterisations of IOPs.


In this paper, we use the Black Sea to highlight the capabilities of this modelling framework. A major output is an upgrade of the BAMHBI model with a three-stream RT module. It is an important development in the context of operational oceanography, as the NEMO-BAMHBI system is used by the Copernicus Marine Service to predict the Black Sea biogeochemistry. However, the Black Sea has many specificities (anoxia, continental shelf) that make it a complex environment to model. The coupled NEMO framework, upgraded with the module presented in this paper, could be transposed to other basins that are close to the Black Sea, such as the Baltic or the Mediterranean Seas. The implementation of RT schemes that allow the simulation of sea surface reflectance is being done on other European seas within Copernicus Marine Service projects with deterministic models. The stochastic version of the RT scheme becomes relevant when it comes to assessing uncertainties. However, other methods are being evaluated such as the use of neural networks to estimate the distribution of IOPs by inverting the RT equation, as in Soto López et al. (2024). A combination of these approaches could be explored in the future, for instance, with this method being used to quantify the sources of uncertainty and ensemble modelling being used to evaluate the propagation of uncertainties in the RT model.



## 6 Conclusion

In this paper, we propose a module to represent marine optics and RT in the NEMO frameworks, based in large part on the model described in Dutkiewicz et al. (2015). As such, it can be coupled to any biogeochemical model that is itself coupled with NEMO. This model simulates three streams of irradiance at high spectral resolution, constituting a tool for simulating in-water irradiance and sea surface reflectance. These variables are particularly relevant for model calibration and validation, as a large amount of both *in situ* and remote-sensing data are available. It also complements the simulation of biogeochemical variables by providing optical quantities that are more closely related to products such as satellite surface chlorophyll.


We use our RT module with the NEMO-BAMHBI modelling framework for the Black Sea, upgrading its capabilities by enabling the simulation of radiometric variables. It is fully coupled with the physical and biogeochemical components of the model, following extensive calibration of the feedback loops for the simulation of temperature and primary production. The inclusion of this new RT scheme does not significantly alter the simulation of physics and biogeochemistry, while providing enriched information on spectral irradiance, which is directly used for the computation of PAR. The comparison with sea surface reflectance from remote-sensing data reveals that the module is able to simulate the main seasonal and spatial patterns in the basin. As such, it complements the biogeochemical variables in providing information on blooms. Some features are still not captured by the model, which leaves room for improvement in our ability to model bio-optics in the Black Sea.





The high spectral resolution of the RT model opens new perspectives in line with algorithms that have been used to derive biogeochemical products from remote-sensing reflectances. The products of surface chlorophyll or suspended particulate matter, for instance, rely on specific wavebands that have to be modelled to truly match model outputs with the datasets that are provided. The recent hyperspectral missions PACE (NASA) and PRISMA (Italian Space Agency) are now providing reflectance data at very high resolution, and models that are able to increase their spectral resolution will also be valuable to make the most of hyperspectral data (Chowdhary et al., 2019). By gathering data on a much larger number of wavebands, new algorithms could be developed to better dissociate the water constituents. More generally, this opens up further possibilities for the assimilation of reflectance data.

*Code and data availability.* The code for the RT model is provided on Zenodo at https://doi.org/10.5281/zenodo.17289633, along with the BAMHBI configuration that has been used for this study (Macé, 2025a). More details on the code organisation can be found in Appendix B. The code for the 1D testing notebook, along with forcing files, can be found on Zenodo at https://doi.org/10.5281/zenodo.17288457 (Macé, 2025b).

Ocean colour data was taken from Black Sea, Bio-Geo-Chemical, L3, daily Satellite Observations (1997-ongoing), E.U. Copernicus Marine Service Information (CMEMS), Marine Data Store (MDS), DOI: 10.48670/moi-00303 (CMEMS, 2025). BGC-Argo data were collated within the Copernicus Marine Service (In Situ) and EMODnet collaboration framework. Data are made freely available by the Copernicus Marine Service and the programmes that contribute to it. DOI: 10.13155/43494 (Copernicus Marine In Situ TAC, 2024). The AERONET-OC data were taken from GSFC NASA AERONET-OC website (accessed 19-09-2024).

## Appendix A: Model calibration using a 1D test model

We first used a 1D test model to calibrate the three-stream RT model that is coupled with the NEMO-BAMHBI system. This light version of the model is fast and efficient computation-wise, which allows to easily compute single profiles of irradiances $E_d$, $E_s$, and $E_u$, as well as sea surface reflectance for a single waveband. This short model does not include any original or significant model development as it is a simple transcription of the existing model presented in Dutkiewicz et al. (2015) into a Jupyter Notebook environment. This notebook is provided with this article as an additional tool that can be used to easily simulate profiles in a simple 1D and non time-dependent framework. The aim was to keep this testing framework as simple as possible. As inputs, it requires the model parameters defined in Table. 1, the absorption and scattering spectra of optically active constituents, and an assumption on phytoplankton composition. By default, it is set as if small flagellates, large flagellates, and diatoms are present in equal proportions. The model then needs BGC-Argo data that include chlorophyll and particle backscattering at 700 nm to derive seawater IOPs. The vertical resolution is also defined in the inputs, along with the wavelengths in which the computation should be performed. The surface radiative forcing is taken here from MAR data for the Black Sea, with a sample of these data for 2018 provided with the files. RT is computed for all the profiles of the float within





the prescribed time range, and results are displayed for sea surface reflectance along BGC-Argo track, and irradiances and IOP profiles.


We used this simple model to simulate profiles based on BGC-Argo data. BGC-Argo floats are drifting buoys that provide both physical and biogeochemical data by collecting profiles every five to ten days. The calibration of the optical properties of phytoplankton and non-algal particles, as well as CDOM absorption was performed using a collection of test profiles ran with this model. IOPs can be derived from measurements of chlorophyll-a, CDOM or backscattering coefficients in available

wavebands (typically 700 nm). Some floats additionally provide profiles of PAR and irradiance in selected wavebands. We can therefore use the surface radiation to run our 1D model and evaluate its ability to reproduce the full measured profiles.

By automatising the simulation of the three streams of irradiance over several wavebands and profiles, we are able to evaluate biases in the model formulation or in the estimation of IOPs. Perturbations can also be added to the IOPs in order to find the

best representation of the optical properties of seawater. By extension, this process could be repeated with data sources other than BGC-Argo data to increase the representativeness and reliability of the calibration. Since the buoys are drifting, we are unable to perform the simulation for the same location at different times. While it allows to cover a larger domain, it offers limited possibility to cover the temporal variability in optical properties.

**Appendix B: Code organisation and model versions**

The up-to-date NEMO-BAMHBI, which does not include the three-stream RT module yet, can be fount on the public Gitlab of the MAST group from the university of Liège (Vandenbulcke and Grailet, 2025). The code for the version of BAMHBI with which the three-stream radiative transfer model has been coupled is provided in the Zenodo archive (see data availability). The code can be found in the `MY_SRC/METEO` directory and is split into three files:

1. `radtrans.F90` hosts the main functions for the computation of RT.

    2. `radtrans_params.F90` hosts parameters for the computation of RT.

    3. `traqsr.F90` hosts the temperature scheme for the coupling with NEMO, as defined in Sect. 2.2.

The parameters for using the RT module must be defined in the `namelist_cfg` file, in the `nam_RADTRANS` namelist. The parameters to be set there are as defined in Sect. 3.3. In addition, the coupling from the RT module towards NEMO

for the computation of temperature has to be activated by setting the flag `ln_qsr_RT = .true.` in the `namtra_qsr` namelist. The `GEO_LR` directory then hosts the absorption and scattering spectra that are necessary to compute RT in the `spectra_water.dat`, `spectra_particles.dat`, `spectra_plankton.dat`, and `cdom_sinusoidal.dat` files. The surface radiative forcing is taken from the MAR configuration, for which all the data are not made available because of



the large size of the dataset.


When the RT module is coupled with BAMHBI, concentrations of the three PFTs and of POC are taken to derive seawater IOPs. Information on water density is taken directly from NEMO to derive CDOM absorption. Then, the irradiance streams are used to compute PAR that is fed back into the biogeochemical model within the `UpdateLight` routine of the `bamhbi.F90` file. In the case where the user only wants to use the three-stream RT module as an observation operator and keep using another RT scheme to derive PAR and temperature, another scheme must be defined in the `bamhbi.h90` file to compute PAR. For physics, another scheme must be chosen (consistent, if possible) in the `namtra_qsr` namelist.

The radiometric outputs are defined in the `traqsr.F90` source file. They can include sea surface reflectance and irradiance streams in relevant wavelengths (i.e. typically those that are measured by satellite sensors of *in situ* stations). In addition, the `ln_radtrans_diags` flag is defined in the `nam_RADTRANS` namelist to output secondary variables such as IOPs or their decomposition by optically active constituent. The output field and file structures must then be defined accordingly for each experiment.

The RT module can also be used with NEMO and without a biogeochemical model. In this case, the `nam_RADTRANS_inputs` namelist is used to provide external data of chlorophyll and POC concentrations. Together with the absorption and scattering spectra, they are used to derive seawater IOPs. In this configuration, the computation of absorption by CDOM remains unchanged and is derived from the seawater density that is computed in NEMO.

*Author contributions.* LM, PB and MG conceptualised the research plan. LV initiated the integration of the radiative transfer model into the NEMO-BAMHBI framework and proposed the NEMO compatible version of the RT module. LM performed the model calibration and validation, the simulations and their analysis. LV and MG provided support with the NEMO-BAMHBI model. JFG ran the MAR configuration used to force the model. JMB and PB provided support with stochastic modelling methods within the NEMO framework. LM wrote the first draft of the paper, PB and MG contributed to the writing and all authors contributed to its review. LM, LV and JFG reviewed the model configuration and code. MG provided funding through the BRIDGE-BS and NECCTON projects.

*Competing interests.* The authors declare that they have no conflict of interest

*Acknowledgements.* This work was funded by the EU H2020 BRIDGE-BS project under grant agreement no. 101000240 and the EU HE NECCTON project under grant agreement no. 101081273. Computational resources have been provided by the Consortium des Équipements de Calcul Intensif (CÉCI), funded by the Fonds de la Recherche Scientifique de Belgique (F.R.S.-FNRS) under Grant No. 2.5020.11 and by the Walloon Region. Part of this research was supported by the Copernicus Service Evolution ODESSA project and the POSYDONIE





project funded by CNES. The authors thank S. Dutkiewicz for sharing the radiative transfer code from the MITGCM-Darwin configuration. We thank the PIs and maintenance staff for their efforts in establishing and maintaining the Gloria and Galata sites. We thank X. Fettweis and the Laboratory of Climatology of the University of Liège for their work on the MAR atmospheric model. We thank P. Lazzari, M. Baklouti, J. Lamouroux and P. Verezemskaya for helpful discussions and suggestions.



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
