# Peer review of "Three-stream modelling of radiative transfer for the simulation of Black Sea biogeochemistry in a NEMO framework"

_EGUsphere, 2025_

## Referee Comment (RC2)

This work presents development and implementation of a radiative transfer model providing high spectral resolution (33 bands) and separate diffuse and direct downwelling irradiance for the underwater light-field. Furthermore, the model used here also simulates the upwelling stream and enables one to estimate the sea surface reflectance. Such development follows the ongoing improvements in ocean optics in the current generation of marine biogeochemistry models, as evidenced by the papers cited by the authors. An extra interesting feature that is presented by the authors is a stochastic perturbation scheme for the IOPs, enabling the authors to analyze the often large uncertainty in this domain. Finally the model was tested and validated with NEMO-BAHMBI in the Black Sea environment.

I think the paper is a nice contribution to the growing literature, but I have a series of comments, mostly minor, that I would like the authors to address, before I can recommend the paper for publication.

**Specific comments:**

- 1. In the Introduction section, the authors list references evidencing the 3D configurations using RT models (around line 35). I would add few other relevant references to the ones cited by the authors:
  - Gregg, W. W., & Rousseaux, C. S. (2016). Directional and spectral irradiance in ocean models: Effects on simulated global phytoplankton, nutrients, and primary production. *Frontiers in Marine Science*, **3**, 240
  - Mobley, C., Sundman, L., Bissett, W., & Cahill, B. (2009). Fast and accurate irradiance calculations for ecosystem models. *Biogeosciences Discussions*, **6**(6), 10,625–10,662.
  - Gregg, W. W., & Casey, N. W. (2007). Modeling coccolithophores in the global oceans. *Deep Sea Research Part II: Topical Studies in Oceanography*, **54**(5-7), 447–477.
  - also the Skakala et al. 2020, 2022 references cited in the paper can be also mentioned in this context.
- 2. Line 65 "RT module ready to be coupled with NEMO" is a little bit confusing, is the module stand-alone? In such case how hard is it to couple it to a range of physics models and how necessarily it has to be NEMO? I would either rephrase, or explain better how much the module is tied specifically to NEMO.
- 3. Line 71 "This model was used in ...." it is a bit confusing which model you say was used in those references, as the two-band model gets easily mixed up with the PISCES model, and even more confusingly, the paragraph seems to be specifically talking about the NEMO physics model. In this regard note that e.g. Ciavatta et al. 2014 used POLCOMS-ERSEM model, so neither NEMO for physics, nor PISCES for biogeochemistry was used.
- 4. A minor thing: in Eq (7), if you are defining R\_below, I would mark z=0 on the right-hand side of the equation as  $z=0_{-}$ , or  $z \to 0_{-}$
- 5. Eq (8-9), I know this is meant to be a general section and then for specific implementation the Q, T, gamma parameter values are listed in Tab.1, but can you link the Table.1 to the text already here, e.g. saying that perhaps representative values of these Q, T, gamma parameters can be found in the Table 1? By doing that, the reader gets a basic quantitative understanding for what are at least the right orders of magnitude for those parameter values...

- 6. Eq. (10) I expect that "T" is temperature, but this needs to be stated. Also please consider that although from the context this might be hopefully clear enough, "T" is the same symbol than used for transmittance in Eq (9), which is never a very good practice...
- 7. Line 200: "POC" often refers to all organic carbon above certain particle size, I suspect you mean here the non-living part of POC, i.e. detritus?
- 8. It is interesting that your attenuation model seems to omit the SPM/sediment, can you comment on this? E.g. in Black Sea environment you don't need to include it near the river mouths/coastlines?
- 9. In Fig.1 the origin of many of the plotted values seems to be undeclared, can you please reference them a bit better? Detritus seems to have some cited references (text around the line 200), but even in this case (and in other cases) it would be good to cite the real origin of these values, e.g. lab experiments, or field studies? Also what are the uncertainties of those values, e.g. I expect the CDOM value to be highly uncertain, can you comment on that?
- 10. Sec 2.4 I know you give references including Mace et al (2025), but introducing perturbation scheme is always non-trivial, perhaps you can include a short paragraph justifying why the specific scheme was chosen (e.g. "first-order autoregressive process"), i.e. what thinking this is based on?
- 11. Sec. 3.1 CDOM it has been derived from Bgc-Argo data as described on the lines 265-270. However, this description I find insufficient is CDOM taken as horizontally spatially varying? If yes, with what effective resolution? It seems that there are some seasonal variations imposed into the CDOM values? With what temporal resolution? I also assume that we are talking about the CDOM forcing based on a seasonal climatology rather than flow-dependent values? Best, can you give a plot showing surface CDOM annual mean concentrations for the forcing and/or Hovmoller plot of horizontally averaged CDOM values (depth x time)? If you have too many Figures, can you put this in the Appendix, or Supporting Information?
- 12. Sec 3.2 lines 280-295 dedicated to the atmospheric forcing of the light module. I would like to see more information on what the atmospheric MAR model exactly simulates when providing the desired outputs, e.g. does it simulate aerosol dynamics and therefore aerosol optical depth? I assume it provides separate outputs for the surface downwelling diffuse and direct irradiance? Also did you validate this model in the Black Sea against observations? Can there be a sentence, or two on how it validates?
- 13. Paragraph around the line 305 I wouldn't say that a forcing bias would not impact reflectance, i.e. reflectance is an AOP and depends on the overall direction of light, so if you had a bias towards e.g. diffuse light, it would influence the reflectance properties..
- 14. Sec 4.1 the paragraph 360-365 makes it sound like PAR is directly compared with the Bgc-Argo, instead what is compared is PAR normalized by its surface value, right? Given the large short-time-scale (sub-daily) PAR variability I wouldn't expect it would be reasonable to compare PAR directly with observations, at least not if standard model (e.g. daily) outputs are used...
- 15. Does Fig.3, the left-hand panel (lambda=380nm) imply that the Argo sees overall higher levels of attenuation than the model? Can this be due to the missing SPM in the model? Can you comment? Curiously why do the data for the shorter wavelengths (the first two panels of Fig.3) near the surface (I guess near the "0" value) start slightly below the black line? It looks like there is some rapid attenuation near the surface in the Bgc Argo data that isn't present in the model? Can you comment on this?

- 16. Small comment to Fig.3-4, can you label the x and y axis differently, i.e. assuming the logarithm has base 10, I would label it as the ratio rather than the log-ratio and use 10^0, 10^-1, 10^-2 (...) on the axes... This means instead of using linear scale for the log, I would use log-scale for the ratios this is generally easier to visually interpret..
- 17. Fig.5: how coarse is the model vertical resolution near the surface? I understand the model has 59 vertical layers unevenly distributed (I assume the model is using z coordinates with fixed depths?), but from the few histogram bins in this Figure it looks that near the surface the water column is vertically not very well resolved. Can you comment on that? Or is the coarse histogram resolution (the x-axis) due to Argo's not measuring across sufficient number of depths? Furthermore, why there are three colors in the Figure when only two runs are compared? You say "blue" and "orange", but I see also a brown color? Also, can you find a way how to make the colors transparent, or show only contours of the histogram bars, so the bars don't get in each other way? Btw. why are only distributions compared? What about pointwise comparison using e,g. RMSE metric?
- 18. Sec 4.2, the text between the lines 380-385: out of curiosity, sorry for my lack of knowledge, but how much can you vary the eta\_h parameter it looks to me like this parameter should be something reasonably constrained?
- 19. When it comes to temperature, why only thermocline depth was compared? Btw. my non-expert understanding of Black Sea is that stratification is dominantly due to salinity and not so much due to temperature? Is thermocline directly associated with pycnocline, or is pycnocline more closely aligned with halocline in the Black Sea? If the latter, it could mean that thermocline is not so important in the Black Sea, or am I getting something wrong?
- 20. Again, as in Fig.3, why you don't show more detailed performance statistics for chlorophyll than the histogram comparison (Fig.6)? Furthermore, similarly to Fig.5, why there are 3 colors (?) and can you make the bars transparent?
- 21. Around the line 420, if you do explain the reflectances by a localized bloom underestimated by the model, please clearly state that you haven't shown this in the paper. Or does the bloom refer to the rChl comparison as per Fig. 10-12? Is it relative to Bgc-Argo? How did you come to the conclusion about the bloom and the reflectance?
- 22. I find very interesting the RMSE difference between rChl and the BAHMBI Chl in Fig.12! I think it would be somehow interesting to show BAHMBI Chl also in Fig.10-11? Can you discuss better this discrepancy between rChl and BAHMBI Chl? Why exactly does the mismatch happen, is it discrepancy in the specific absorption, or scattering coefficients between the satellite and the ecosystem model? One conclusion one could make based on this discrepancy is the high uncertainty in chl (and perhaps also Rrs?) comparison when validating the model with the satellite!
- 23. In Section 4.6 how were the perturbations standard deviations selected? I hear that e.g. CDOM can have very high uncertainty in its specific absorption coefficient, is 50% standard deviation enough? Why?
- 24. Discussion section I'd say it reads a lot like a Summary section, rather than Discussion. Maybe you want to change the title to reflect upon that?
- 25. Discussion on the lines 550-560 I'd say that comparing Rrs has the advantage of capturing all the optically active tracers across its spectra. However, if there is large uncertainty in the specific absorption, or scattering coefficients (i.e. mismatch between the satellite and the ecosystem model), the comparison with satellite can be a bit

- arbitrary, and I believe this is regardless of whether it is done through using inversion, or via model-derived Rrs, no?
- 26. The mention of Rrs assimilation on the lines 565-570 is an interesting point. One thing that comes to my mind is that due to the relatively complex relationship between Rrs and the biogeochemistry model state variables (including inversion), the Rrs DA might be easier to do for DA approaches that attempt to directly represent cross-covariances between variables in the background covariance matrix (like EnKF and similar). For variational (3DVar) methods that often simplify the background covariance matrix (e.g. as part of parametrizing it), Rrs assimilation might pose a bigger challenge do you have any comments on that?